# Effects of Rhizobia Isolated from Coffee Fields in the High Jungle Peruvian Region, Tested on *Phaseolus vulgaris* L. var. Red Kidney

**DOI:** 10.3390/microorganisms10040823

**Published:** 2022-04-15

**Authors:** Jesus Lirio-Paredes, Katty Ogata-Gutiérrez, Doris Zúñiga-Dávila

**Affiliations:** Laboratorio de Ecología Microbiana y Biotecnología, Department of Biology, Faculty of Science, Universidad Nacional Agraria La Molina, Lima 15024, Peru; jesusliriopar@gmail.com (J.L.-P.); kogata@lamolina.edu.pe (K.O.-G.)

**Keywords:** symbiosis, *Rhizobium*, yield, *Phaseolus vulgaris*, functional nodules

## Abstract

Soils in the high jungle region of Peru continuously face erosion due to heavy rain, which leads to significant nutrient losses. Leguminous plants may provide a sustainable solution to this problem due to their ability to fix atmospheric nitrogen with the help of symbiotic rhizospheric microbes that reside in their root nodules and help restore soil fertility. The aim of this study was to isolate native rhizobial strains that can form functional nodules in red kidney beans to help improve their growth, development, and yield in field conditions. *Rhizobium* strains were isolated from soil samples collected from coffee fields using bean plants as trap hosts. The strain RZC12 was selected because it showed good root nodule promotion and a number of PGPR (plant-growth-promoting rhizobacteria) attributes. In the field, bean plants inoculated with the strain RZC12 and co-cultivated with coffee plants produced approximately 21 nodules per plant, whereas control plants produced an average of 1 nodule each. The inoculation with RZC12 significantly increased plant length (72.7%), number of leaves (58.8%), fresh shoot weight (85.5%), dry shoot weight (78%), fresh root weight (85.7%), and dry root weight (82.5%), compared with the control. The dry pod weight produced by the plants inoculated with RZC12 was 3.8 g, whereas the control plants produced 2.36 g of pods. In conclusion, RZC12 is a promising strain that can be used in field conditions to improve the overall productivity of red kidney beans.

## 1. Introduction

Over the last 50 years, the human population has rapidly increased, placing enormous pressure on the agricultural sector to increase its productivity. It is estimated that, by the year 2050, 70% more (from the current levels) food grains and other food products will be needed to feed the entire global population [1]. Recently, farmers have widely adapted the use of fertilizers and other agrochemical products to overcome soil nutritional deficiencies, meet the increasing demand for food products, and increase their revenue [2,3]. Among the various soil nutrients needed for optimum plant growth, nitrogen is considered to be the most important and most limiting. Proper supply management and practices for efficient usage of this particular macronutrient are necessary to improve crop productivity and yield, especially in agricultural fields suffering from its deficiency [1,4,5]. However, the extensive use of chemical fertilizers to achieve high agricultural productivity is a major contributor to soil, water, and atmospheric pollution. As is widely acknowledged, the most important challenge of this century is to increase the productivity and yield of major staple crops in an environmentally friendly, sustainable way with minimum water wastage, while utilizing the resources at our disposal to their maximum potential [6,7,8].

Soil erosion due to excessive rainfall is a very common natural phenomenon in the high-altitude forests of Peru. This phenomenon causes soil impoverishment and subsequently leads to reduced soil fertility. Sustainable agriculture, therefore, is a much-needed approach. This challenge gives agricultural lands an opportunity to recover from nutrient losses and helps restore soil fertility. Growing legumes improves overall soil quality and increases fertility through biological nitrogen fixation [9,10]. The major atmospheric-nitrogen-fixating organisms known are of the genus *Rhizobium*. The members of this genus live in a symbiotic association with plants and play an important role in increasing crop yield, especially in nitrogen-deficient soil. *Rhizobium*–legume symbiosis has been extensively studied and explored to improve soil fertility. This association benefits the legume crops and the crops co-cultivated with them [11,12,13,14]. Microbes from the genus *Rhizobium* are known to have the ability to capture atmospheric nitrogen by reducing it enzymatically into ammonia within the root nodules and make it readily available to the plant for their growth and development. Some rhizobia are also reported to exhibit plant-growth-promoting features, such as inorganic phosphate solubilization, indole acetic acid (IAA), cytokinins, gibberellins, and iron transportation, among others.

*Phaseolus vulgaris* var. red kidney was introduced into Peru by the Bean Program in the 1960s after extensive field trials were carried out in the Urubamba Valley in Cusco, Peru. This plant species is now being cultivated in more Central American countries, suggesting that it has adapted to warm climates [15,16]. At the same time, coffee is among the most important economic crops primarily cultivated in the high jungle region of Peru. Coffee is a long-term plant, and it only begins to generate profits for farmers after three years of cultivation, when the berries are produced. In this context, the cultivation of *Phaseolus vulgaris* var. red kidney bean inoculated with native strains of *Rhizobium*, grown together with coffee plants, gives farmers a sustainable alternative to increase their income, while helping to overcome soil nitrogen deficiency. The aim of our study was to isolate and characterize native *Rhizobium* strains from the functional nodules of *P. vulgaris* var. red kidney bean plants cultivated along with coffee plants in the Chanchamayo region of Peru and to develop them as potential bio-inoculants to increase the yield of bean plants.

## 2. Materials and Methods

### 2.1. Soil Sampling

Soil samples were collected from the coffee fields located in the Peruvian region known as “Selva Alta” in areas originally occupied by tropical premontane (submontane, foothill) forests. Two zones were sampled: the Genova-IRD Selva station in Chanchamayo (11°5′41″ S, 75°21′50″ W) at 1200 m above sea level and the San Jose station, Villa Rica, in the center of San Miguel de Eneñas, Oxapampa, Pasco (10°44′49″ S, 75°12′44″ W), located 1484 m above sea level (Appendix A). The annual average precipitation in Chanchamayo is 2000 mm, and the maximum and minimum temperatures are 30.1 °C and 16.7 °C, respectively, with relative humidity ranging from 70% to 95%. Villa Rica has an average annual rainfall of 1978 mm. Its maximum temperature is 26.8 °C, and the minimum temperature is 10.6 °C, with a relative humidity ranging from 55% to 90%. Five healthy, average-sized coffee plants were collected at each location. Each plant was completely extracted with 1 kg of soil adjacent to its root. Soil samples for physicochemical analysis were sent to the testing laboratory for soils, plants, water, and fertilizers at Universidad Nacional Agraria La Molina. The analysis samples were air-dried and sieved in a 2 mm mesh sieve. Sand, silt, and clay percentage were determined using the hydrometer method. Soil salinity was determined by measuring the electrical conductivity of the liquid extract in a 1:1 soil/water ratio. The pH was established in a ratio 1:1 (soil:water) using a potentiometer. The common gas–volumetric method was used to determine carbonate (CaCO_3_). The organic matter (OM) was measured by the Walkley and Black method (potassium dichromate oxidation). The available phosphorous was extracted from the soil with 0.5 M NaHCO_3_ solution adjusted to pH 8.5 using a procedure modified from Olsen method. The K+ exchangeable base was determined by the extraction with NH_4_OAC adjusted to pH 7.0 followed by quantification using atomic absorption [17].

### 2.2. Rhizobia Isolation

Coffee rhizospheric and non-rhizospheric soil samples (Table 1) were used to isolate *Rhizobium* strains with *P. vulgaris* var. red kidney bean as a trap host. Surface-sterilized seeds were sown in pots containing sampled soils and maintained at 25 °C. After 25 d, plant growth parameters, including height, fresh and dry weight of root and aerial parts, number of flower buds, and number of nodules, were evaluated. Root nodules were separated from the roots, dehydrated in sterile microtubes using silica gel, and stored from 2 weeks at 5 °C [18].

For rhizobia isolation, nodules were hydrated for 30 min in distilled water. Then, they were disinfected by soaking them in alcohol 70% for 1 min and in 3% sodium hypochlorite for 3 min. Finally, the nodules were rinsed 5 times in sterile distilled water. Then, nodules were crushed in 100 μL of 0.85% saline solution 8. Suspensions were spread-plated on CRYEM (Congo red yeast extract mannitol) agar [19], and the plates were incubated at 28 °C for 10 d. Colonies of rhizobia were identified by their typical appearance (i.e., white, glistening, elevated, showing entire margins, and low or no absorption of Congo red) [19,20]. Isolates were individually analyzed by Gram staining, purified by sub-culturing, and cryopreserved in 16% *v*/*v* glycerol–YEM broth at −80 °C [21].

### 2.3. Phosphate Solubilization

Strains were grown to OD_600_ = 0.3 (~10^8^ CFU/mL) in the YEM (yeast extract mannitol) broth. A 5 µL drop of each strain culture was spotted onto plates of NBRIP medium [22] and on the same medium with tricalcium phosphate (Ca_3_(PO_4_)_2_) substituted by the same quantity of dicalcium phosphate (CaHPO_4_) or hydroxyapatite (Ca_5_(PO_4_)_3_(OH)). Plates were incubated at 28 °C for 15 d. Colony and halo diameters were measured, and the results are expressed in terms of RSE (relative solubilization efficiency) determined by the following formula: solubilization diameter/growth diameter × 100% [23]. The assay was repeated twice with three replicates.

### 2.4. Indole Acetic Acid Production

IAA (indole acetic acid) production was estimated colorimetrically using Salkowski reagent [24,25]. Strains were grown in 3 mL of the YEM broth [18] supplemented with 5 mM L-tryptophan for 48 h at 28 °C. Each bacterial concentration was adjusted to 10^8^ CFU/mL. A 500 μL aliquot of each culture was taken and centrifuged at 12,000 rpm for 5 min. Supernatants were transferred to sterile test tubes, mixed with Salkowski reagent (1:4), and incubated for 30 min in the dark at room temperature. The appearance of a reddish color was considered a positive result and was measured at 530 nm. IAA concentration was calculated from an adjusted calibration curve with a maximum concentration of 50 ppm. The experiment was repeated twice with three replicates.

### 2.5. Rhizobia Authentication

The ability of the strains to infect plants of the original host and to form functional nodules was determined. Seeds were surface-sterilized as described earlier and allowed to germinate at 25 °C in sterile Petri dishes containing a wet filter paper [18]. After 2 d, seedlings were aseptically transferred to tubes with polypropylene beads supplemented with a nitrogen-free nutrient solution [26]. The positive control only was supplemented with a nutrient solution with nitrogen. Rhizobial strains were inoculated in treatment seedlings. The negative control was inoculated with the YEM broth without bacteria, and the positive control was supplemented with a nutrient solution containing 0.05% of KNO_3_ as a nitrogen source. Plants were harvested after 25 d of growth. The agronomic parameters of the fresh and dry weight of shoots, roots, and pods were recorded. The numbers of leaves, flowers, pods, and nodules were also documented. Pink-colored root nodules were considered functional nitrogen-fixing nodules. The experiment used five plants per treatment.

### 2.6. Plant Assays under Controlled Conditions

Based upon the previous tests, two bacterial strains were selected for further plant assays. Strains were grown on YEM agar plates at 28 °C for 48 h. Inoculum was prepared by collecting bacterial growth from the Petri dishes and suspending it in a 0.85% saline solution. The concentration was adjusted until a concentration of 10^8^ CFU/mL was obtained. Seeds were disinfected as described earlier and soaked in the bacterial inoculum solution for 30 min. Control seeds were exposed to the saline solution (0.85%) for the same length of time. Seeds were allowed to germinate in Petri dishes as described earlier. Seedlings were transferred to pots with sterile sand and maintained at controlled conditions of temperature (24 °C), humidity (80%), and a photoperiod of 16 h of light and 8 h of darkness, for 50 d. After 50 d, the plants were harvested, and agronomical parameters, such as shoot and root length and weight, were evaluated. The experiment was repeated twice with ten plant replicates per treatment.

### 2.7. Molecular Characterization

Genomic DNA from strain RZC12 was extracted with an AxyPrep Bacterial Genomic DNA Miniprep Kit (Axygen Scientific, USA) in accordance with the manufacturer’s instructions. The PCR amplification of 16S ribosomal RNA was performed using primers fD1: (5′-CCGAATTCGTCGACAACAGAGTTTGATCCTGGCTCAG-3′) and rD1: (5′-CCCGGGATCCAAGCTTAAGGAGGTGATCCAGCC-3′) [27]. The program used for PCR amplification included the following steps: initial denaturation at 94 °C for 3 min, followed by 30 cycles at 94 °C for 1 min, annealing at 55 °C for 45 s, and extension at 72 °C for 1.5 min, and then final extension at 72 °C for 5 min using an Eppendorf thermocycler (Eppendorf AG, Hamburg, Germany). The amplified fragments were purified with the AxyPrep PCR Cleanup Kit (Corning Life Sciences, Tewksbury, MA, USA) in accordance with the manufacturer’s instructions and subsequently sequenced by a commercial service (Macrogen Inc., Seoul, Korea). The strain sequence was identified through a search of the NCBI database [28]. Related sequences were retrieved from the Genbank database, and a multiple alignment was generated using Clustal X2 [29]. A phylogenetic analysis was performed using the neighbor-joining method using the software MEGA 11 [30].

### 2.8. Field Trial

A selected strain was evaluated in a field trial. This strain was grown in the YEM broth for 48 h until a concentration of ~10^8^ CFU/mL was obtained. Bean seeds were uniformly mixed with the inoculum, dried in the shade for 30 min, and planted in rows located between *Coffea arabica* plants. The experiment was set up in a field plot of 14 × 8 m^2^ and with a 1.5 m configuration of row space. Coffee plants were planted at 1 m^2^ spacing and co-cultivated with common bean var. Red kidney. The bean experiment comprised 4 rows. A total of 20 plants were sown per row considering that two bean seeds were sown per hole at a spacing of 30 cm. RZC12 and control treatments were sown in alternating rows. The control treatment used seeds mixed with the sterile YEM broth. Bean plants were harvested 60 d after sowing. Three plants per row were uprooted from the plot for agronomic characteristics’ evaluation, such as plant height, pod number, and fresh and dry weight of aerial and root portions. The experiment was performed using 12 replicates per treatment.

### 2.9. Statistical Analysis

Statistical analysis was performed using the Statgraphics Centurion XIX software. Data were analyzed using analysis of variance (ANOVA). Where the F values were significant, comparisons of means analysis was applied using the least significance test (LSD) at the 0.05 probability level (*p* < 0.05).

## 3. Results

### 3.1. Soil Analysis

A3 and B3 samples belong to the same coffee field, while A1 and B1 correspond to an adjoining coffee field. Additionally, C is another field close to the other ones but without plants. All of them share the same coordinates because of their proximity. The physicochemical characterization of Chanchamayo soil samples revealed that they were moderately acidic, with pH values ranging from 4 to 5, had a moderate level of organic matter, and were of a sandy loam texture. The conductivity values showed that samples B1 and A3 had very low salinity, whereas samples C and B3 had slightly higher values. By contrast, the soil samples collected from Villa Rica were strongly acidic, with a pH of 3.5, and were non-saline, with a loam texture and a moderate level of organic matter (Table 1).

Beans used as trap plants in B1 soil samples (Table 2) showed higher numbers of functional nodules than plants grown in other soils (Figure 1).

### 3.2. Isolation of Rhizobia

Most nodules recovered from soils A3 and B3 were non-functional. Plants grown in soil samples A2 and B2, collected in Villa Rica, were devoid of nodules. Bean plants cultivated in soil B1 had relatively higher root and shoot lengths, fresh and dry weights, and numbers of leaves and flower buds than plants grown in the other soils, in agreement with their nodulation profiles.

Only samples from Chanchamayo were selected for rhizobia isolation because they developed effective pink nodules compared with samples from Villa Rica. A total of 11 strains were isolated from the nitrogen-fixing root nodules obtained in the plant trap assays. All isolates were Gram-negative. Colony morphology and color on CRYEM agar plates were non-uniform; the color varied from cream to light pink, whereas the colony size varied from 1 mm to 4.5 mm after 3 d of growth (Figure 2). All strains produced mucoid colonies with high amounts of exopolysaccharides.

### 3.3. Plant Growth Promotion Ability

The results of PGPR activities are summarized in Table 3. The qualitative analysis of phosphate solubilization showed that all strains, except RZC10, could solubilize dicalcium phosphate, tricalcium phosphate, and hydroxyapatite. RZC10 failed to produce a halo in the presence of tricalcium phosphate as a substrate. Strains were better adapted to solubilizing hydroxyapatite in comparison with the other phosphate sources. All strains could produce IAA at high concentrations. Among all the isolates, RZC12 and RZC17 produced IAA in large amounts: 48.9 and 41.4 ppm, respectively.

### 3.4. Authentication Assays

All 11 strains could nodulate bean plants (Table 4); however, most nodules were found to be non-functional. Strains RZC12, RZC13, and RZC17 could produce functional nodules. Inoculation with strain RZC12 induced the maximum formation number of functional nodules compared with other strains (Figure 3).

Plant growth parameters, such as shoot and root fresh weight, showed significant increases in plants inoculated with the tested strains in comparison with the negative control. The maximum root fresh weight (1.28 g) and root and shoot dry weight (0.105 and 0.281 g, respectively) were recorded in the case of inoculation with RZC12. RZC12 was able to increase fresh root weight (0.105 g) compared with the control (0.061 g). StrainsRZC2, RZC3, RZC5, RZC6, RZC12, RZC17, RZC18, and the control (N+) displayed better effects on dry aerial weight compared with the non-inoculated N-control (Table 4).

### 3.5. Plant Assay under Laboratory Conditions

Based on the IAA production, phosphate solubilization, and nitrogen-fixing nodule formation results, two strains were selected to evaluate their effects on bean plants under controlled conditions. The inoculation resulted in a significant increase in aerial length and functional nodule formation (Table 5). Shoot length with RZC12 and RZC17 were 41.77 and 40.04 cm, respectively, and it was 34.7 cm in the control. RZC12 was able to induce nodule formation better than RZC17. It also had higher shoot and root dry weight compared with the control.

### 3.6. Field Experiment

RZC12 was selected for a field experiment based on the in vitro and in vivo results. The results clearly demonstrate that RZC12 can improve various agronomic parameters of *Phaseolus vulgaris* var. red kidney bean (Figure 4) when it is cultivated alongside coffee plants. Significant increases in shoot length, leaf number, and nodule number were observed in plants inoculated with the *Rhizobium* strain in comparison with the control (Table 6).

Moreover, aerial fresh weight, fresh root weight, shoot dry weight, and root dry weight showed increases of 85.5%, 78%, 85.7%, and 56.9%, respectively, in comparison with the control plants. Significant differences in bean dry pod weight (3.8 g/plant) were also observed as compared with that of the control (2.36 g/plant).

### 3.7. Phylogenetic Analysis of the Strain

A comparison of the 16S rRNA gene of the RZC12 strain against type strains of bacterial species recorded in the NCBI database showed a 99.71% identity with *Rhizobium vallis* NR_116835.1. The strain 16S rRNA sequence had a length of 1362 bp and was deposited in NCBI database under accession number MN607599. The phylogenetic tree shows that strain RZC12 was also clustered with *Rhizobium vallis* (Figure 5).

## 4. Discussion

Legumes can form a positive symbiotic relationship with nitrogen-fixing soil bacteria called rhizobia. The rhizobium–legume is a well-known model system that requires a complex signal exchange between both organisms [31]. Functional nitrogen-fixing root nodules are commonly red or pink. The term “functional” here denotes their ability to actively fix nitrogen for plant usage. The characteristic red or pink color appears due to the presence of leghemoglobin. Nitrogenase is another important enzyme for nitrogen fixation activity and the sustainability of the symbiotic association between the plant and rhizobia, but it is extremely sensitive to environmental oxygen [32,33,34,35]. Leghemoglobin provides root nodules with the ability to protect the nitrogenase from oxygen. Sometimes, this kind of symbiosis does not show good results in the absence of functional nodules. Furthermore, nodulation failure in acidic soil conditions is a common phenomenon and is observed in soils with pH values less than 5 [36]. This finding explains the failure of root nodulation in plants cultivated in soil samples A2 and B2 (Villa Rica), which have very low pH values. Low soil pH was found to reduce nodule numbers on legumes such as common bean, lentil, pea, and soybean by more than 90% and nodule dry weight by more than 50%. However, some legumes species, such as *Lupinus* spp. and *Mimosa* spp., were found to exhibit nodulation under acidic soil conditions [37]. On the other hand, plants sown in rhizospheric soil B1 showed better growth and development compared to the others. This may be explained by the high number of functional nodules found in their roots compared to the plants sown in the other soil samples. Nodules’ development is influenced by soil pH and organic matter content. Some rhizobia strains are also known to secrete certain plant growth hormones, such as IAA, which has a positive effect on plant growth and development. IAA also plays an important role in the formation and development of root nodules. In addition, *Rhizobium* species with phosphate solubilization capability can also effectively release phosphorous, another important macronutrient for plants, from complex inorganic and organic compound pools and therefore have the potential to improve plant yield and reduce fertilizer requirements [38]. Rhizobia strains with the abilities to produce IAA and solubilize phosphate can improve the growth and quality of a variety of crops [39]. These PGPR attributes are important in intercalated crops, such as the ones included in this study. All strains in the present study could solubilize phosphate and secrete IAA. Among the strains tested, RZC12 and RZC17 showed the best ability to produce IAA (48.9 and 41.4 ppm, respectively).

The efficacy of the rhizobial strains infecting nitrogen-fixing nodules was determined using the authentication test. Finding other non-rhizobial endophytes inside the root nodules is not uncommon; therefore, plant authentication assays were carried out to distinguish these from the rhizobial strains [40,41]. Non-nodulating rhizobia could also be found in the nodules. These rhizobia probably lost one of the symbiotic plasmid-containing genes necessary for their ability of root nodulation [42]. All eleven strains isolated in this study showed positive results in the authentication tests and so were assigned to the genus *Rhizobia*. Sometimes, symbiotic associations between the plant and rhizobia remain ineffective, with no significant benefit for the plant. A greater production of nodules means a greater productivity of the plant, but not in all cases. In a study carried out on beans, the inoculation of *R. tropici* gave a higher nodule dry weight compared to *R. ethyli*, while this last strain promoted a significant increase in the dry weight of pods [43]. The capacity for N_2_ fixation can be assessed by comparing yields of inoculated plants with the +N controls as well as with the commercial inoculant strains. Strains can be ranked by comparing yield as a percentage of that achieved by the +N treatment or by the best strain or by the commercial strain, as required [44]. Numerous studies have shown the positive effects of inoculating native rhizobia strains in legumes compared to uninoculated controls under field conditions [45]. In our study, the majority of the strains showed an increase in shoot and root weights compared with the control. The most effective strain was RZC12, which showed a significant increase in fresh root (105%) and shoot (96%) weight and a tendency to improve root (127%) and shoot (100%) dry weight. Positive results were shown in a similar experiment using *Rhizobium* spp. and *Bacillus* spp. strains as inoculants. Growth parameters, such as plant height, fresh and dry aerial weight, and number of flower buds, resulted in a significantly high value for common bean grain yield, compared to the untreated control [46]

In vivo plant experiments conducted under laboratory conditions showed that strains RZC12 and RZC17 could significantly increase the plant shoot length, which could be linked with their ability to fix atmospheric nitrogen and other PGPR traits [44]. Among the two strains, RZC12 could induce the formation of the maximum number of root nodules and exhibited better PGPR abilities, which steered us to select this strain for further field studies. Rhizobium strains are not only N-fixers in symbiosis with legumes but are also major promoters of plant growth due to their production of phytohormones and solubilization of phosphates [47].

Numerous studies have linked the nitrogen-fixing activity of rhizobia strains with the improvement in plant yields, measured as an increase in the production of biomass dry weight [48,49]. This study highlights the effectiveness of RZC12 to nodulate red kidney bean plants and significantly increase most of the agronomical parameters evaluated, including shoot/root fresh and dry weight, numbers of pods per plant, and dry pod weight per plant. These results are not surprising because it is widely known that symbiotic root rhizobia have the ability to fix high amounts of atmospheric nitrogen and make it readily available to plants for their growth requirements [50]. Moreover, in the case of legume crops, pods are also considered marketable products for the farmers and so are included in the estimations of overall bean crop yield [50,51]. This study demonstrated that the application of RZC12 can improve the yield of red kidney bean crops and can be used as a part of the biological management of soil fields linked to co-cultures or rotation systems.

## 5. Conclusions

The rhizobial authentication assay demonstrated that rhizobia strains did not develop in the strongly acidic soils of Villa Rica. In vitro and in vivo results of rhizobial effectivity obtained in the laboratory were further validated in the field. RZC12 was found to be an effective root-nodulating rhizobia strain, with the ability to promote plant growth of beans. Improvement in agronomic growth parameters, such the fresh and dry weight of plants and pods, is an important indicator for crop yield and productivity. In this study, strain RZC12 demonstrated its ability to increase crop yield by almost doubling the production of pods in field conditions.

## Figures and Tables

**Figure 1 microorganisms-10-00823-f001:**
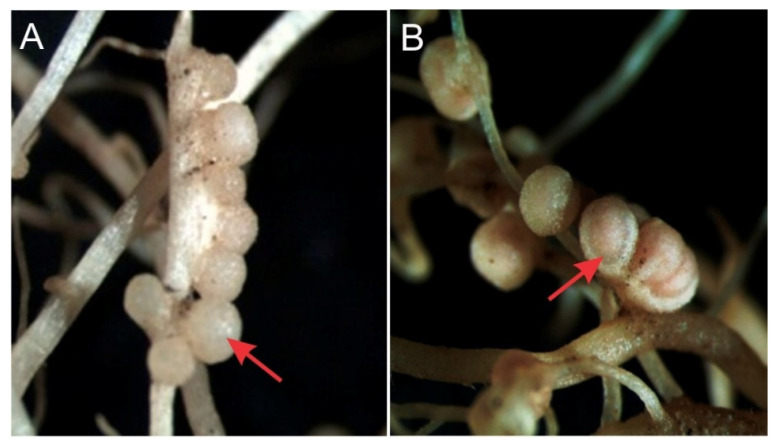
**Red arrows point** (**A**) Non-functional nodules generated in soil B3. (**B**) Functional nodules in soil B1 (40× magnification).

**Figure 2 microorganisms-10-00823-f002:**
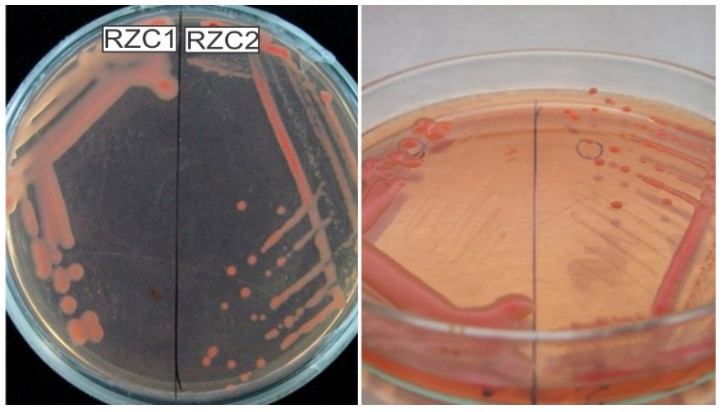
(**Left**): Isolation plate where the strain RZC1 is clearly differentiated from the strain RZC2. (**Right**): Convex elevation and pink color of rhizobial strains.

**Figure 3 microorganisms-10-00823-f003:**
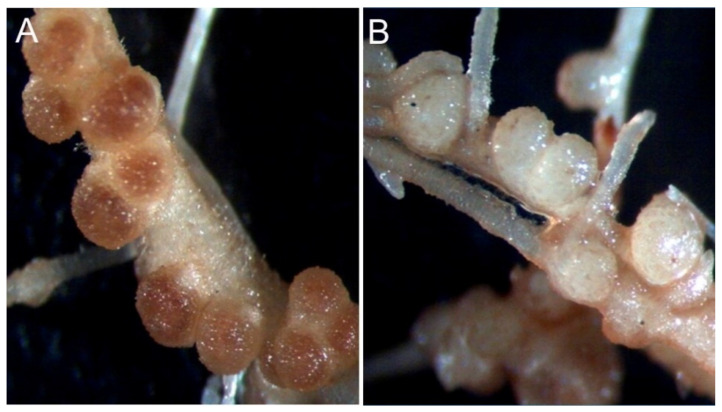
(**A**) Functional nodules produced by strain RZC12. (**B**) Non-functional nodules produced by the RZC1 strain.

**Figure 4 microorganisms-10-00823-f004:**
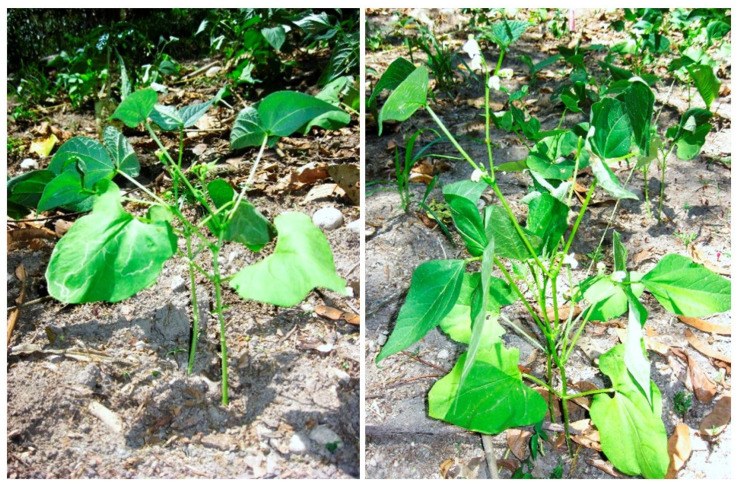
Red kidney beans planted in an intercrop, non-inoculated control (**left**) and inoculated with the RZC12 strain (**right**).

**Figure 5 microorganisms-10-00823-f005:**
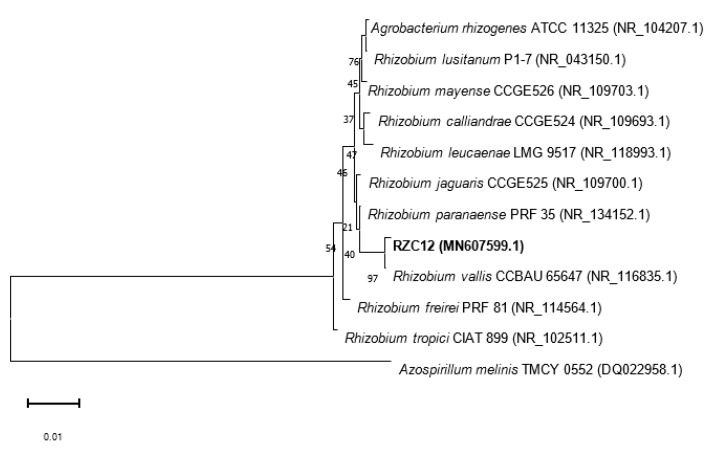
Phylogenetic tree of sequences of the 16S rRNA gene (1354 positions) of *Rhizobium* sp. The phylogenetic reconstruction method of neighbor joining, and the distances were calculated in accordance with the Kimura two-parameter model. The values at the branch points indicate bootstrap support (1000 pseudoreplicates; only values of 50% or above are shown). The tested *Rhizobium* strain is indicated in bold. *Agrobacterium tumefaciens* was used as outgroup to root the tree. Scale bar, 1 nt substitution per 100 nt.

**Table 1 microorganisms-10-00823-t001:** Soil characteristics.

Soil	Kind of Soil	Zone	pH (1:1)	E.C. (1:1) dS/m	O.M.%	N% 5% of O.M.	P ppm	K ppm	Texture	Description
A2	Non-rhizospheric	Villa Rica	3.51	0.53	3.64	0.182	5.5	95	Loam	Very strongly acidic and non-saline soil, moderate organic matter content
A3	Non-rhizospheric	Chanchamayo	5.76	0.33	2.19	0.1095	6.1	133	Sandy loam	Moderately acidic and non-saline soil, moderate organic matter content
B1	Rhizospheric	Chanchamayo	4.74	0.77	1.86	0.093	6.5	113	Sandy loam	Strongly acidic and non-saline soil, low organic matter content
B2	Rhizospheric	Villa Rica	3.43	0.86	3.91	0.1955	28	108	Loam	Very strongly acidic and non-saline soil, moderate organic matter content
B3	Rhizospheric	Chanchamayo	5.75	4.78	3.31	0.1655	25	1323	Sandy loam	Moderately acidic and strongly saline soil, moderate organic matter content
C	Uncultivated	Chanchamayo	6.5	0.96	3.24	0.162	28	160	Sandy loam	Weakly acidic and weakly saline soil, moderate organic matter content

E.C.: electric conductivity, O.M.; organic matter.

**Table 2 microorganisms-10-00823-t002:** Growth parameters of red kidney beans used as a trap host.

Soil	Pink		White		Root		Shoot		Root Fresh		Shoot Fresh		Root Dry		Shoot Dry		N°		Floral	
	Nodules		Nodules		Length (cm)		Length (cm)		Weight (g)		Weight (g)		Weight (g)		Weight (g)		Leaves		Buttons	
A2	0	c	0	c	12.83 ± 3.31	b	38.83 ± 5.38	bc	0.69 ± 0.25	bc	3.01 ± 0.55	bc	0.068 ± 0.03	a	0.275 ± 0.06	bc	2.67 ± 0.52	b	2 ± 1.67	bc
A3	3.67 ± 2.25	b	7.5 ± 3.62	a	14.17 ± 2.02	ab	41.5 ± 9.85	ab	0.87 ± 0.16	ab	3.07 ± 0.7	bc	0.068 ± 0.02	a	0.262 ± 0.04	bc	3.33 ± 0.52	ab	2.83 ± 2.56	bc
B1	14.67 ± 8.51	a	6.67 ± 4.16	ab	14.5 ± 0.5	ab	52.33 ± 8.15	a	1.09 ± 0.41	a	4.24 ± 1.53	a	0.077 ± 0.02	a	0.42 ± 0.16	a	3.67 ± 0.58	a	2.0817	a
B2	0	bc	0	c	8.33 ± 0.76	c	26.83 ± 3.55	c	0.5 ± 0.02	c	2.45 ± 0.08	c	0.04	b	0.217 ± 0.01	c	2.67 ± 0.58	b	1 ± 1	c
B3	3 ± 2.97	bc	11.5 ± 5.39	a	16.17 ± 1.6	a	44.83 ± 8.95	ab	0.96 ± 0.15	a	3.85 ± 0.7	ab	0.07 ± 0.06	a	0.217 ± 0.06	abc	3.33 ± 0.52	ab	4.5 ± 1.76	ab
C	0	c	2.17 ± 2.48	bc	14.42 ± 3.02	ab	45.5 ± 11.93	ab	0.98 ± 0.2	a	3.98 ± 0.69	a	0.063 ± 0.02	ab	0.217 ± 0.08	abc	3.83 ± 0.75	a	3.67 ± 2.58	abc

Means followed by the same letter in the columns do not differ significantly (*p* ≤ 0.05) according to LSD test.

**Table 3 microorganisms-10-00823-t003:** PGPR activities of rhizobia strains isolated from soils samples collected from Chanchamayo.

Strain	Dicalcium Phosphate Solubilization	Tricalcium Phosphate Solubilization	Hydroxyapatite Phosphate Solubilization	IAA (ppm)
	RSE ^a^	RSE	RSE	
RZC1	36% ± 2.51	17% ± 1.3	70% ± 2.37 abc	30.4 ± 0.45
RZC2	20% ± 4.31	10% ± 1.86	33% ± 2.99 bcd	30.6 ± 0.16
RZC3	27% ± 2.32	15% ± 2.6	55% ± 3.71 d	18.9 ± 0.46
RZC4	20% ± 1.91	17% ± 1.51	30% ± 6.48 bcd	25.4 ± 0.29
RZC5	25% ± 1.84	17% ± 3.91	40% ± 5.54 a	30.3 ± 0.14
RZC6	36% ± 4.72	17% ± 0.57	44% ± 6.27 bc	38.4 ± 0.09
RZC10	40% ± 2.23	0%	30% ± 0.72 a	20.2 ± 0.15
RZC12	33% ± 3.33	15% ± 0.55	45% ± 2.28 cd	48.9 ± 0.21
RZC13	30% ± 2.41	9% ± 2.02	33% ± 4.18 bcd	33.9 ± 0.36
RZC17	23% ± 2.7	15% ± 1.24	23% ± 1.89 ab	41.4 ± 0.44
RZC18	23% ± 0.4	15% ± 3.9	42% ± 4.84 cd	23.6 ± 0.2

^a^ RSE: relative solubilization efficiency. Means followed by the same letter in the columns do not differ significantly (*p* ≤ 0.05) according to LSD test.

**Table 4 microorganisms-10-00823-t004:** Authentication effect of rhizobia strains on red kidney beans.

Treat.	N° White Nodules		N° Pink Nodules		Root Length (cm)	Shoot Length (cm)		N° Leaves	Root Fresh Weight (g)		Shoot Fresh Weight (g)		Root Dry Weight (g)		Shoot Dry Weight (g)	
**Control**	0	d	0	d	13.3 ± 3.84	24.9 ± 3.12	ab	4 ± 0.58	0.808 ± 0.06	d	1.295 ± 0.13	d	0.061 ± 0.02	c	0.15 ± 0.03	b
**N+**	0	d	0	d	13.8 ± 1.94	26.9 ± 0.33	a	4	1.228 ± 0.25	ab	2.295 ± 0.36	a	0.083 ± 0.015	b	0.281 ± 0.07	a
**RZC1**	17.3 ± 6.11	a	2 ± 1.91	bcd	11.8 ± 1.72	26.7 ± 4.7	a	3	1.15 ± 0.13	abc	1.773 ± 0.43	bcd	0.081 ± 0.013	bc	0.225 ± 0.03	ab
**RZC2**	10.5 ± 3.42	ab	1 ± 0.82	cd	12.2 ± 1.5	25.3 ± 1.43	ab	3 ± 0.5	1.098 ± 0.22	abc	1.998 ± 0.4	abc	0.082 ± 0.01	b	0.261 ± 0.06	a
**RZC3**	14.2 ± 5.40	ab	0	d	13.2 ± 2.81	24.7 ± 2.4	ab	3 ± 0.89	1.018 ± 0.1	c	1.838 ± 0.47	bcd	0.082 ± 0.01	b	0.234 ± 0.08	a
**RZC4**	5 ± 3.92	cd	0	d	13.8 ± 1.45	22.2 ± 2.89	b	3 ± 0.96	1.098 ± 0.27	abc	1.72 ± 0.33	cd	0.083 ± 0.04	b	0.212 ± 0.07	ab
**RZC5**	7.2 ± 7.36	bcd	0	d	13.7 ± 2.55	26.8 ± 3.04	a	3 ± 1	0.994 ± 0.2	cd	2.004 ± 0.61	d	0.073 ± 0.02	bc	0.257 ± 0.1	a
**RZC6**	18 ± 10.15	a	1 ± 1.73	cd	11.8 ± 0.29	26.7 ± 0.85	a	4 ± 0.58	1.153 ± 0.23	abc	2.03 ± 1.98	abc	0.081 ± 1.98	bc	0.262 ± 0.05	a
**RZC10**	13.3 ± 5.51	cd	0.7 ± 1.16	cd	13.2 ± 1.02	26.2 ± 2.47	a	3	1.02 ± 0.14	bcd	1.83 ± 0.33	abc	0.075 ± 0.004	bc	0.207 ± 0.01	ab
**RZC12**	18 ± 4.24	abc	10.5 ± 8.89	a	13.7 ± 3.54	26.8 ± 1.49	a	4	1.288 ± 0.15	a	2.213 ± 0.30	abc	0.105 ± 0.02	a	0.281 ± 0.08	a
**RZC13**	7.8 ± 3.22	bcd	4.5 ± 6.61	bcd	13.5 ± 1.31	25.2 ± 3.6	ab	4 ± 0.5	1.18 ± 0.15	abc	1.855 ± 0.11	abc	0.082 ± 0.01	b	0.227 ± 0.02	ab
**RZC17**	15.8 ± 11.01	a	5.2 ± 1	b	13.3 ± 1.5	26.9 ± 3.29	a	3 ± 0.55	1.262 ± 0.21	a	2.108 ± 0.37	abc	0.083 ± 0.02	b	0.272 ± 0.08	a
**RZC18**	10.3 ± 8.08	abc	0.7 ± 0.58	cd	12.3 ± 2.8	26.7 ± 3.05	a	3 ± 1.16	1.233 ± 0.14	ab	2.187 ± 0.26	abc	0.081 ± 0.02	bc	0.273 ± 0.045	a

Means followed by the same letter in the columns do not differ significantly (*p* ≤ 0.05) according to LSD test.

**Table 5 microorganisms-10-00823-t005:** Effect of rhizobia strain inoculation on the growth and root nodulation of red kidney beans under laboratory conditions.

Treatment	Shoot Length (cm)		Root Length (cm)	N° Leaves	N° Flowers	N° Pots	N° Nodules		Shoot Fresh Weight (g)	Root Fresh Weight (g)	Shoot Dry Weight (g)	Root Dry Weight (g)
RZC12	41.77 ± 4.92	a	22.63 ± 3.88	5.86 ± 0.69	3 ± 0.82	1.43 ± 0.53	26.43 ± 7.72	a	4.04 ± 0.65	2.5 ± 0.48	2.42 ± 0.56	0.28 ± 0.06
RZC17	40.04 ± 4.05	a	20.36 ± 4.4	60.82 ± 4.39	3.29 ± 0.76	1.43 ± 0.53	18.71 ± 5.59	b	3.85 ± 0.29	2.53 ± 047	2.72 ± 0.48	0.25 ± 0.03
CONTROL	34.7 ± 3.02	b	19.27 ± 3.02	5.57 ± 0.53	3 ± 0.82	1.29 ± 0.49	1.57 ± 1.27	c	3.32 ± 0.6	2.19 ± 0.41	2.24 ± 0.46	0.23 ± 0.03

Means followed by the same letter in the columns do not differ significantly (*p* ≤ 0.05) according to LSD test.

**Table 6 microorganisms-10-00823-t006:** Effect of inoculation with Rhizobium-RZC12 on the growth and root nodulation of red kidney beans in field studies.

Treat.	Shoot Length (cm)		Root Length (cm)	N° Leaves		N° Flowers	N° Nodules		Shoot Fresh Weight (g)		Root Fresh Weight (g)		Pod Fresh Weight (g)		Shoot Dry Weight (g)		Root Dry Weight (g)		Pod Dry Weight (g)	
RZC12	33.59 ± 4.34	a	15.31 ± 3.23	5.4 ± 0.97	a	3.3 ± 0.68	21 ± 8.68	a	8.44 ± 2.48	a	1.62 ± 0.51	a	5.68 ± 0.92	a	4.27 ± 2.48	a	0.182 ± 1.3	a	3.8 ± 0.67	a
CONTROL	19.45 ± 1.85	b	13.5 ± 3.58	3.4 ± 0.7	b	2.8 ± 0.79	0.7 ± 1.06	b	4.55 ± 1.15	b	0.91 ± 0.17	b	4.16 ± 1.23	b	2.3 ± 1.15	b	0.116 ± 0.58	b	2.36 ± 0.63	b

Values followed by the same letter in the columns do not differ significantly (*p* ≤ 0.05) according to LSD test.

## Data Availability

Data will be made available upon request to the corresponding author: dzuniga@lamolina.edu.pe.

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
