# Peer review of "Effects of Rhizobia Isolated from Coffee Fields in the High Jungle Peruvian Region, Tested on Phaseolus vulgaris L. var. Red Kidney"

_microorganisms, 2022, doi:10.3390/microorganisms10040823_

Round 1

Reviewer 1 Report

In this work, authros isolate native rhizobial strains from acidic soils under the influence of coffee plants. After a polyphasic characterization, that included ability of plant colonization, rate of plant growth enhacement, production of functional nodules and release of IAA, they select one strain that has been further characterized. This strain has been tested in field trials, where it produced a significant increase of plant growth.

In general, the paper is well written and structured. I only have several comments that try to improve the text and the experimental analysis:

1) Molecular characterization:

The PCR amplification of 16S ribosomal RNA was performed using primers 143

fD1 and rD1, which were used in a previous work. fD1 and rD1 amplify not only amplifies Rhizobium but most eubacteria. In case that the strain was not axenic, a percentage of PCR amplification bands from other bacteria that might “contaminate” the sequences. This might introduce “artificial” point mutations. The best procedure for the identification of the Rhizobial strain is to clone the PCR fragments in a plasmid and then sequence several ones. This procedure will allow the identification of contaminant bacteria (in case they were there) and to ensure that the mutations in the sequence are real. Please, provide information of the sequences that are obtained by this procedure.

2) Sample collection points and soil analysis:

A figure (in supplementary material) indicating the exact points in which soil samples were collected is desirable. Authors should explain, for instance, the difference between samples A3 and B3. Were taken at different locations? The same for the “uncultivated” sample.

Regarding the soil physicochemical analysis, please provide the methodology for the determinations. In this context, it does not make sense to include CaCO3 %, since is 0 in all the cases. But it would be desirable to know about the content in nitrogen (Nitrate and ammonium) in the different types of samples, since this is a factor that might influence occurrence of Rhizobia in the soil and nodulation in the plants.

3) I do not understand why beans are cultivated between the coffee plants. Do authors have any indication about Rhizobia richness under the influence of coffee plants?

4) PGPR activities (Table 3):

Authors show P solubilization activity and IAA production. Regarding the quantification of IAA, authors should include normalized values (ppm/mg protein or similar) and the standard deviation. This normalization is valued to compare the IAA production activity with other known Rhizobial strains.

It would be desirable to show the N2 fixation activity of the strains, since this activity is essential for sustaining plant growth. Nitrogenase activity is a standardized method that can be measured under anoxic conditions.

Minor points:

Tables 1 and 2: to make them easier to read and interpret, they should use the same order in the soil samples. The same applies for tables 3 and 4.

For all the tables: please, indicate the standard deviation of the values provided, and the number of the measurements done. Indicate the meaning of “a” and “b” after the numbers.

3.4. Plant Assay Under Laboratory Conditions: indicate what is “under controlled conditions” in the text to facilitate reading.

Reference 26: DOI is not correct. Replace by https://doi.org/10.1128/jb.173.2.697-703.1991

Author Response

Please see the attachment (in green)

Reviewer 2 Report

Presented research article describes the isolation of rhizobia strains that can form functional nodules in red kidney beans. Several strain were selected and one of them was further validated in field conditions. Plants inoculated with this strain demonstrated improved crop yield and productivity.

Although the presented manuscript is well written authors should stress their attention on several points.

1. Is not clear how many plants were used in the reported experiments and how many replicates were done. Without these details the results cannot be considered as reliable.

2. All data presented in Tables 2, 4, 5, 6 are without standard deviation or standard error! Please correct!

3. In the same tables there are some letters (a, b, c, d) but their meaning is unclear.

4. It is true that functional nitrogen-fixing nodules are commonly red or pink, but how authors will explain that plants without functional nodules (for example these inoculated with RZC18 strain) demonstrate the same phenotype as the plants with red nodules?

From the other side, plants inoculated with strain RZC6 posses only one functional nodule but the same phenotype as well nodulated plants inoculated with RZC12. Probably nodules obtained with strain RZC6 have higher nitrogenase activity and in this way RZC6 can be even better candidate for further analyses? Authors should perform additional analyses to study also the nitrogenase activity (by acetylene reduction assay for example) in the nodules.

5. In the discussion (rows 291-292) authors said “Plant–rhizobia symbiosis is considered effective when, in laboratory and field experiments, inoculated plants gain 75% of the positive control (N+) weight. But in the presented study all data are compared only to the negative (N-) control. Please corect.

5. Authors should try to avoid in the conclusion very general statements (row 316-317)

Some minor points:

1. Authors must describe all the abbreviation used in the text, including PGP and PGPR.

2. Why the strains presented in Table 4 are not in numerical order?

3. To my point of view the title of the paper is inappropriate. This title did not describe well the work presented in the manuscript and is different from the aim of the study reported in the abstract.

Author Response

Please see the attachment (in green)

Reviewer 3 Report

The manuscript needs to be corrected as noted in the attached file.

Author Response

Please see the attachment in green

Reviewer 4 Report

The research topic of the paper, titled “Effect of Rhizobia Inoculation on Phaseolus vulgaris L. var. Red Kidney Bean Yield in a Coffee Bean Co-cultivation in the High Jungle Region of Peru” by Lirio-Paredes Jesus et al., is very interesting but, the presentation did not meet expectations. Additionally, this paper could be made more effective by improving its experimental and field sections. Please consider in detail and respond to the given suggestions.

Page 2, lines 54-57: Summarisation of common PGP characteristics of rhizobia (such as production of IAA, phosphate solubilisation, ACC deaminase and siderophore production) should be briefly described.

Page 2, lines 85-86: All of the analysed soil parameters and the methods used, should be added.

Page 2, lines 89-90: How many soil samples were used and were they rhizospheric or non-rhizospheric?

Page 2, line 93: What was the storage temperature, and how long were nodules stored in these conditions?

Page 3, line 94: Nodule surface sterilisation method needs to be described.  

Page 3, lines 101-107: How many replications were used?

Page 3, lines 108-115: How many replications were used? For IAA determination you should use a minimum of 3 replication (6 replications are optimal).

Page 3, line 115: The maximum concentration of the IAA calibration curve may be added.  

Page 3, line 118:  Nodule surface sterilisation method needs to be described. 

Page 3, line 122: For the positive control specify what was the concentration of nitrogen was used.

Page 3, line 133: What were the concentrations of the inoculum (CFU/ml)? Since more isolates have been used for treatments, the initial inoculum concentrations for all strains should be unformed (determined according to the OD600 nm).

Page 4, line 144: Although the reference is provided, sequence of the used primers should be added.

Page 4, line 150-152: NCBI may be used instead of EzTaxon (BLAST function) https://blast.ncbi.nlm.nih.gov/Blast.cgi?PROGRAM=blastn&PAGE_TYPE=BlastSearch&LINK_LOC=blasthome. Also, the accession numbers of strains from the database which were used for the phylogenetic analysis.

Page 4, line 153: Which program was used for the construction of phylogenetic tree? If raw sequences were  processed (end-trimmed) what was the program used?

Page 4, Section 2.8. „Field Trial“ line 154-161: Description of the experiment: is it a combined crop of beans and coffee (we conclude that it is from the title)? What were the setting of the combined crop (in rows, between rows) and the distance in the row and between the rows? More information should be given about the experiment itself, whether in addition to inoculation, some mineral or organic fertilizer was used to provide the required amounts of major nutrients. Second, give the Latin name for the coffee. Third, was the experiment set up in the soil where no legumes have been grown before?  Why are the agronomic parameters monitored only for beans and not for the coffee plant, which is included in the combined crop (will the PGRP properties of rhizobium affect coffee if they are planted as the combined crop?). If you have not followed all the parameters for the coffee crop (although it would be interesting to see if it was also affected by PGPR), you may need to consider redefining the title of the paper. Consider changing the headline into: Effects of Rhizobia Isolated from Coffee Fields in the Peruvian Region to Phaseolus Vulgaris L. Var. Red 2 Kidney Bean Yield. 

Line 159: Was any positive control included (with a recommended dosage of chemical fertiliser or commercially available rhizobia-based fertiliser)?

Before Section 3. Results: Description of used statistical analysis is missing. Please add the paragraph describing it.  

Page 4, 3.1: Subheading “Isolation of Rhizobia” should be renamed to “Soil analysis” and isolation of bacteria should be described in a separate paragraph (“Isolation of Rhizobia”).  

Page 6, Line 174-175 Table 2.: Consider including standard deviation.  

Page 6, Line 179: Did you conclude that root nodules were unfunctional based on their color only, or some additional growth parameters were considered? Please specify in more detail.

Page 6, Line 183: Results should be described in more detail: was there any connection between recorded growth parameters and soil quality and state statistical differences or similarities.  

Page 6, Line 185: As you also done Gram staining, please include it in the Materials and Methods section.  

Page 6, Line 187: Figure 2 is missing.  

Page 7, Table 3: These results should also be described by statistics so the best isolates could be stand out more effectively.  Also, the formula for RSE calculation should be included in the Materials and Methods section.

Page 7, Table 4: After adding the description of used statistical analysis (see comment for Before Section 3. Results) you should define the statistical differences between strains and include them in the tables.  

Page 8, Line 218: Add comparison between strains and positive control, especially for SDW.

Page 9, Line 247: Consider using NCBI base. If NCBI is used it shows similarity of 99.78% with MW041263.1 (Rhizobium vallis).

Page 9, Line 249: Add that your strain was deposited in NCBI under accession number MN607599, and include the length of deposited fragment.

Page 10, Figure 5: Consider using different species as outgroup (from Bacillus, Pseudomonas or Serratia genus) as they are more genetically distant from your stain than Agrobacterium.

Page 10, Section 4. Discussion: The discussion needs to be majorly revised: add references that can be compared with your results; add one introductive paragraph stating which plant species could be improved by rhizobial inoculation, and then add references regarded to beans and rhizobial inoculation in order to compare results. Add more research regarding rhizobial inoculation in acid soils. Add some references to compare growth parameters of different plant species when inoculated by rhizobia-based inoculation.

In order to better explain what you should do, I recommend checking the following papers:

https://doi.org/10.1016/j.rhisph.2022.100487

https://doi.org/10.1007/s42729-020-00171-8

https://doi.org/10.1038/s41598-021-83235-8

Author Response

The manuscript, please see the attachment (the changes are in red)

REVIEWER 4

Page 2, lines 54-57: Summarisation of common PGP characteristics of rhizobia (such as production of IAA, phosphate solubilisation, ACC deaminase and siderophore production) should be briefly described.

DONE (lines 59-61)

Page 2, lines 85-86: All of the analysed soil parameters and the methods used, should be added.

DONE (lines 89-100)

Page 2, lines 89-90: How many soil samples were used and were they rhizospheric or non-rhizospheric?

The reference of the table was added. Table 1 (6 samples)

Page 2, line 93: What was the storage temperature, and how long were nodules stored in these conditions?

DONE (lines 107-108)

Page 3, line 94: Nodule surface sterilisation method needs to be described.

DONE (lines 109-113)

Page 3, lines 101-107: How many replications were used?

DONE, line 127

Page 3, lines 108-115: How many replications were used? For IAA determination you should use a minimum of 3 replication (6 replications are optimal).

DONE, line 137

Page 3, line 115: The maximum concentration of the IAA calibration curve may be added. 

This data was added. Line 137.

Page 3, line 118:  Nodule surface sterilisation method needs to be described. 

Instructions were included. Lines 109-113.

Page 3, line 122: For the positive control specify what was the concentration of nitrogen was used.

DONE, line 146.

Page 3, line 133: What were the concentrations of the inoculum (CFU/ml)? Since more isolates have been used for treatments, the initial inoculum concentrations for all strains should be unformed (determined according to the OD600 nm).

DONE, line 131.

Page 4, line 144: Although the reference is provided, sequence of the used primers should be added.

DONE (lines 170-171)

Page 4, line 150-152: NCBI may be used instead of EzTaxon (BLAST function) https://blast.ncbi.nlm.nih.gov/Blast.cgi?PROGRAM=blastn&PAGE_TYPE=BlastSearch&LINK_LOC=blasthome. Also, the accession numbers of strains from the database which were used for the phylogenetic analysis.

DONE (line 179, 317-319)

Page 4, line 153: Which program was used for the construction of phylogenetic tree? If raw sequences were processed (end-trimmed) what was the program used?

The information has been completed, line 181-182.

Page 4, Section 2.8. „Field Trial “line 154-161: Description of the experiment: is it a combined crop of beans and coffee (we conclude that it is from the title)?

Yes, but the aim of this work was only to evaluate the effect of rhizobia inoculation on beans, that is why the title was changed.

What were the setting of the combined crop (in rows, between rows) and the distance in the row and between the rows?

DONE, lines 187-198

More information should be given about the experiment itself, whether in addition to inoculation, some mineral or organic fertilizer was used to provide the required amounts of major nutrients.

No mineral or organic fertilizer was added, only the rhizobia inoculate

Second, give the Latin name for the coffee. Third, was the experiment set up in the soil where no legumes have been grown before?  

Latin name (line 188)

That was a coffee field, no legumes were sown before

Why are the agronomic parameters monitored only for beans and not for the coffee plant, which is included in the combined crop (will the PGRP properties of rhizobium affect coffee if they are planted as the combined crop?).

We suggest co-cultivation of coffee and beans, since coffee plants take a long time to reach their productive stage (3 years). Usually, in that time, many farmers invest their resources with no economic retributions. Phaseolus vulgaris var. Red kidney is a legume that is sown in the jungle and could be used to feed farmers and their families, while coffee plants are growing. Using a legume as a co-culture system associated to coffee plants, would give many advantages to the soil in long-term due to the biological nitrogen fixation that could improve field productivity.

If you have not followed all the parameters for the coffee crop (although it would be interesting to see if it was also affected by PGPR), you may need to consider redefining the title of the paper. Consider changing the headline into: Effects of Rhizobia Isolated from Coffee Fields in the Peruvian Region to Phaseolus Vulgaris L. Var. Red 2 Kidney Bean Yield. 

The title has been changed according to your suggestion

Line 159: Was any positive control included (with a recommended dosage of chemical fertiliser or commercially available rhizobia-based fertiliser)?

Only negative control without inoculation was included in the assay

Before Section 3. Results: Description of used statistical analysis is missing. Please add the paragraph describing it.  

Statistical análisis description was added (lines 199-203)

Page 4, 3.1: Subheading “Isolation of Rhizobia” should be renamed to “Soil analysis” and isolation of bacteria should be described in a separate paragraph (“Isolation of Rhizobia”).  

DONE

Page 6, Line 174-175 Table 2.: Consider including standard deviation.

The standard deviation was included on table 2

Page 6, Line 179: Did you conclude that root nodules were unfunctional based on their color only, or some additional growth parameters were considered? Please specify in more detail.

Only color was taking in count

Page 6, Line 183: Results should be described in more detail: was there any connection between recorded growth parameters and soil quality and state statistical differences or similarities.

The connection was explained on discussion (lines 365-369)

Page 6, Line 185: As you also done Gram staining, please include it in the Materials and Methods section.  

It was added (row 116-117)

Page 6, Line 187: Figure 2 is missing.  

Figure was included 

Page 7, Table 3: These results should also be described by statistics so the best isolates could be stand out more effectively.

Done, table 3

Also, the formula for RSE calculation should be included in the Materials and Methods section.

Formula was added (lines 126-127)

Page 7, Table 4: After adding the description of used statistical analysis (see comment for Before Section 3. Results) you should define the statistical differences between strains and include them in the tables.  

Done, table 4

Page 8, Line 218: Add comparison between strains and positive control, especially for SDW.

A reference of the statistical results was added (lines 281-283).

Page 9, Line 247: Consider using NCBI base. If NCBI is used it shows similarity of 99.78% with MW041263.1 (Rhizobium vallis).

Done (lines 317-319)

Page 9, Line 249: Add that your strain was deposited in NCBI under accession number MN607599, and include the length of deposited fragment.

Done (lines 317-319)

Page 10, Figure 5: Consider using different species as outgroup (from BacillusPseudomonas or Serratia genus) as they are more genetically distant from your stain than Agrobacterium.

Done. Figure 5 was reformulated

Page 10, Section 4. Discussion: The discussion needs to be majorly revised: add references that can be compared with your results; add one introductive paragraph stating which plant species could be improved by rhizobial inoculation, and then add references regarded to beans and rhizobial inoculation in order to compare results. Add more research regarding rhizobial inoculation in acid soils. Add some references to compare growth parameters of different plant species when inoculated by rhizobia-based inoculation.

Done. The information has been included on the Discussion section (lines 349-350, 362-369, 390-401, 415-417).

References were added (references section)

Round 2

Reviewer 1 Report

In the updated manuscript all my concerns were addressed. Nice work.

Reviewer 2 Report

The revised version of the manuscript is significantly improved. In addition, authors provided the required information concerning experiments design and statistic.

As a minor point,  in row 180-183 the same sentence is repeated twice. Please correct.

Reviewer 4 Report

Thank you for revising the manuscript carefully according to the suggestions. There is adequate background information provided in the manuscript. Now, the research methods and results are presented clearly.